# Automated Definition of Skeletal Disease Burden in Metastatic Prostate Carcinoma: A 3D Analysis of SPECT/CT Images

**DOI:** 10.3390/cancers11060869

**Published:** 2019-06-21

**Authors:** Francesco Fiz, Helmut Dittmann, Cristina Campi, Matthias Weissinger, Samine Sahbai, Matthias Reimold, Arnulf Stenzl, Michele Piana, Gianmario Sambuceti, Christian la Fougère

**Affiliations:** 1Department of Nuclear Medicine and Clinical Molecular Imaging, University Hospital Tübingen, 72076 Tübingen, Germany; helmut.dittmann@med.uni-tuebingen.de (H.D.); matthias.weissinger@med.uni-tuebingen.de (M.W.); Matthias.Reimold@med.uni-tuebingen.de (M.R.); christian.lafougere@med.uni-tuebingen.de (C.l.F.); 2Department of Medicine-DIMED, Nuclear Medicine Unit, University Hospital of Padua, 35128 Padua, Italy; cristina.campi@gmail.com; 3Nuclear Medicine Unit, Hospital Universitaire Henri Mondor, 94010 Créteil, France; sam.sahbai@gmail.com; 4Department of Urology, University Hospital Tübingen, 72076 Tübingen, Germany; urologie@med.uni-tuebingen.de; 5Department of Mathematics, University of Genoa, 16146 Genoa, Italy; piana@dima.unige.it; 6Department of Health Sciences, Nuclear Medicine Unit, University of Genoa, 16146 Genoa, Italy; sambuceti@unige.it; 7iFIT Cluster of Excellence, University of Tübingen, 72076 Tübingen, Germany

**Keywords:** mCRPC, SPECT/CT, Computer-assisted diagnosis, XOFIGO, Therapy response assessment

## Abstract

To meet the current need for skeletal tumor-load estimation in castration-resistant prostate cancer (CRPC), we developed a novel approach based on adaptive bone segmentation. In this study, we compared the program output with existing estimates and with the radiological outcome. Seventy-six whole-body single-photon emission computed tomographies/x-ray computed tomography with 3,3-diphosphono-1,2-propanedicarboxylic acid from mCRPC patients were analyzed. The software identified the whole skeletal volume (S_Vol_) and classified the voxels metastases (M_Vol_) or normal bone (B_Vol_). S_Vol_ was compared with the estimation of a commercial software. M_Vol_ was compared with manual assessment and with prostate specific antigen (PSA) levels. Counts/voxel were extracted from M_Vol_ and B_Vol_. After six cycles of ^223^RaCl2-therapy every patient was re-evaluated as having progressive disease (PD), stable disease (SD), or a partial response (PR). S_Vol_ correlated with that of the commercial software (R = 0.99, *p* < 0.001). M_Vol_ correlated with the manually-counted lesions (R = 0.61, *p* < 0.001) and PSA (R = 0.46, *p* < 0.01). PD had a lower counts/voxel in M_Vol_ than PR/SD (715 ± 190 vs. 975 ± 215 and 1058 ± 255, *p* < 0.05 and *p* < 0.01) and B_Vol_ (PD 275 ± 60, PR 515 ± 188 and SD 528 ± 162 counts/voxel, *p* < 0.001). Segmentation-based tumor load correlated with radiological/laboratory indices. Uptake was linked with the clinical outcome, suggesting that metastases in PD patients have a lower affinity for bone-seeking radionuclides and might benefit less from bone-targeted radioisotope therapies.

## 1. Introduction

Castration-resistant prostate cancer (CRPC) is defined by rising prostate specific antigen (PSA) levels under androgen blockade and by the eventual diffuse metastatic spread [1,2,3]. In these patients, skeletal metastases can represent the most relevant prognostic factor, by impairing the static function of the skeleton and by reducing the available space for hematopoiesis [4,5,6].

Diffuse skeletal CRPC metastatization, once considered a terminal diagnosis, can today be managed by many different approaches, including new-generations of taxanes, second-line hormonal therapy, and radioisotope treatments [7,8,9,10,11,12]. As these medications might have varying effectiveness and cause different side effects, the choice of therapy sequence usually requires multidisciplinary disease-management. Nevertheless, the most effective sequence of systemic treatments is still a matter of discussion and patient-specific components are likely to play a relevant role [13,14,15].

Being able to assess treatment response reliably is a pre-requisite for therapy selection and sequencing. Response evaluation may be performed either by analyzing tumor marker blood levels or by serial imaging. Serum PSA level is the most used marker, but alkaline phosphatase might also be helpful in assessing metastasis-dependent bone turnover [16,17].

Circulating markers are, however, dependent on the degree of tumor differentiation and can be altered by concomitant therapies [18,19,20]. On the other hand, evaluating medical imaging, whether morphological or radioisotope-based, can be challenging in the presence of a high number of metastases or in case of therapy-related changes, such as the “flare” phenomenon [21].

Obtaining an automated evaluation of the skeletal tumor burden is one of the greatest current unmet clinical needs; in recent years, an increasing number of software applications have been developed for this purpose [22]. Existing applications are mostly based on automated thresholding, using standardized uptake value on positron emission tomography (PET) or count values on bone scans; in this setting, telling apart metastases-related uptake from other non-malignant sources of increased bone turnover can be challenging.

To improve the reliability of metastases detection and to obtain a reliable estimation of tumor load, we developed a specific computational tool based on segmentation analysis. This algorithm uses the computed tomography (CT) information to identify and segment all hyperdense localizations within the skeletal system automatically to define the overall metastatic bone compartment. In a second step, information from the co-registered single-photon emission computed tomography (SPECT) or PET images can be extracted for this volume. In this study, we tested this approach on a series of CRPC patients and validated the analysis against different clinical parameters.

## 2. Results

### 2.1. Volumetric Assessment and Comparison between Systems

The estimate of total osseous tissue (skeletal volume S_Vol_, sum of metastases volume M_Vol_, trabecular volume B_Vol_, and cortical volume C_Vol_) showed a tight concordance between our software (EXCALIBUR, University of Genoa, Genoa, Italy) and the commercial application. Mean global skeletal volume was in fact 3875 ± 1513 mL and 3881 ± 1499 mL as measured by the commercial software application and by our computational program, respectively (R = 0.99, *p* < 0.001). Mean counts/voxel were 439 ± 71 and 435 ± 61, respectively, with an R correlation index of 0.92 (*p* < 0.001, data not shown). Mean M_Vol_ was 362 ± 249 mL (range 85–1194 mL), corresponding to 27 ± 20% of the total trabecular bone. 

The majority of tumor burden was located within the axial skeleton and in the hipbones (M_Vol_ 335 ± 140 mL); 61 patients (81%) had skeletal localizations within the appendicular long bones (M_Vol_ 32 ± 19 mL).

### 2.2. Volume Characteristics

Mean Hounsfield density was comparable for M_Vol_ (590 ± 136) and C_Vol_ (531 ± 92) but was significantly lower for B_Vol_ (251 ± 78, *p* < 0.001). Higher tracer activity was measured within M_Vol_ (939 ± 279 mean counts/voxel), as compared to B_Vol_ (462 ± 196 mean counts/voxel, *p* < 0.001). Activity concentration within C_Vol_ was even lower (271 ± 106 mean counts/voxel), see Table 1. Mean counts/voxel were directly correlated to mean Hounsfielf Units (HU) for both M_Vol_ (R = 0.52, *p* < 0.01) and B_Vol_ (R = 0.74, *p* < 0.001, Figure 1).

### 2.3. Tumor Volume, Number of Lesions, and PSA Level

The software based semi-automatic lesion identification (M_Vol_) and the manual lesion count showed a tight correlation, for the whole skeletal system (R = 0.61, *p* < 0.001, Figure 1) and the axial segments (R = 0.64, *p* < 0.001), but not for the appendicular ones (R = 0.07, *p* = 0.52). Likewise, the number of manually counted lesions correlated with the percent of invasion of the trabecular bone by metastases (INV%) within the whole (R = 0.68) as well as axial (R = 0.69) skeleton, *p* < 0.001, Figure 1.

Remarkably, PSA level as a marker of tumor load correlated with our measures of bone involvement (M_Vol_ R = 0.46, *p* < 0.01; number of manually counted lesions R = 0.67, *p* < 0.001; mean HU of M_Vol_ R = 0.42, *p* < 0.01; mean HU B_Vol_ R = 0.52, *p* < 0.001; and M_Vol_/B_Vol_ ratio R = 0.75, *p* < 0.001). 

### 2.4. Impact of Superscan Status

Twelve patients (16%) were classified as superscan. CRPC patients with a superscan had a higher mean HU than non-superscan patients in B_Vol_ (*p* < 0.001 Figure 2) as well as in M_Vol_ (*p* < 0.01, Figure 2). 

Mean counts/voxel in M_Vol_ were higher in superscan subjects than in non-superscan (1104 ± 291 vs. 886 ± 251, *p* < 0.05, Figure 2); conversely, mean counts/voxel in B_Vol_ were not significantly different between the two subpopulations.

Finally, superscan was associated with a higher INV% of trabecular bone by osteoblastic lesions (M_Vol_/B_Vol_ ratio: 40 ± 23% vs. 21 ± 16%, *p* < 0.01, Figure 5), which was reflected, at qualitative analysis, by a higher number of visually observable metastases (Number of metastases = 106 ± 22 vs. 61 ± 23, *p* < 0.001). See Table 1 for a detailed analysis.

### 2.5. Therapy Response Assessment

According to the response assessment criteria, which are further detailed in the Material and Methods, 21 patients (28%) were classified with progressive disease (PD), while 35 subjects (46%) were classified with stable disease (SD), and 20 patients (26%) with a partial response (PR).

Patients who presented a PD status after the therapy completion exhibited a markedly lower activity in the M_Vol_ (715 ± 190 counts) in the pre-therapy scan, when compared to those with PR (N = 20, 975 ± 215 counts, *p* < 0.05) or SD (N = 35, 1058 ± 255 counts, *p* < 0.01). Of note, similar findings were found within B_Vol_, where patients with PD displayed the lowest activity (PD 275 ± 60, PR 515 ± 188 and SD 528 ± 162 counts, *p* < 0.001). See Figure 3.

At receiver operating characteristics (ROC) analysis, both M_Vol_ and B_Vol_ mean counts/voxel were predictive of therapy effectiveness (M_Vol_ 0.895 and B_Vol_ 0.943 for, p < 0.001). The best threshold value of mean counts/voxel for discriminating patients with progressive disease was in fact 805 in the M_Vol_ (sensitivity 84%, specificity 81%) and 385 in the B_Vol_ (sensitivity 84%, specificity 100%). No differences were observed in absolute volume of metastases (M_Vol_) across the therapy outcome groups. See Figure 4.

Patients presenting a superscan pattern of radioactivity distribution were evenly distributed among the three groups (PD: 3/21 or 14%, SD 6/35 or 17%, and PR 3/20 15%, *p* = non significant.

PSA levels showed considerable variations during the therapy. On average, starting from staging to the end of the therapy, it increased by 135 ± 99% in PD patients and by 27 ± 81% in PR subjects. Conversely, PSA decreased by 40 ± 16% in SD patients. However, due to the marked spread of the PSA-level course among patients, no statistically significant difference could be demonstrated among these groups.

See Table 2 for a detailed analysis.

## 3. Discussion

The present paper describes a computational approach to the problem of skeletal tumor burden quantification by means of SPECT/CT data. The robustness of the bone recognition was testified by the tight correlation between the total bone volume as detected by our approach and by a commercial application. The automated identification of bone tumor volume could not be compared with a reference standard, as, to the best of our knowledge, a commercially available CT-based tumor burden estimator does not yet exist. However, the estimates provided by this software tool show a tight concordance with the traditional measures of tumor burden performed with imaging and blood tests.

In our analysis, we considered both the raw figure of tumor volume as well as the percent of invasion, in other words, the ratio between tumor and trabecular volume. Both estimators strongly correlated with the number of manually counted lesions as well as with the PSA values.

As expected, patients with a superscan status at planar bone scans presented a higher tumor volume as well as a higher percent of trabecular bone invaded by bone metastases. These patients presented also a higher bone metastases density and counting rate; however, mean counts in trabecular bone did not significantly differ from those in the trabecular bone of non-superscan patients. 

However, the distribution of radioactivity into the metastases, as well as into the trabecular bone, appears to play an important role in the therapeutic effectiveness of ^223^RaCl_2_. Previous studies have in fact shown that the distribution of the bone scan tracers mirrors that of ^223^RaCl_2_ [23,24]. In our population, patients who were found to have a disease progression at the end of the therapy presented a lower counting rate at the baseline SPECT/CT not only within the known tumor lesions, but also in the trabecular bone. It might be hypothesized that a lower counting rate at SPECT/CT could correspond to a lesser ^223^RaCl_2_ avidity and thus to a reduced absorbed dose. Thus, one might hypothesize that micrometastases (if present) in trabecular bone exhibiting only a low counting rate will receive an insufficient therapeutic dose. It is worth noting that tumor volume was conversely and not significantly different across patients with partial response, stable, or progressive disease. As a consequence, semi-quantitative evaluation of bone tracer uptake at baseline might be useful for ^223^RaCl_2_ therapy stratification.

Patients presenting with a superscan finding were equally distributed in the three response groups, in other words, the presence of a superscan was not per se associated with imaging-based therapy response in our population. Actually, the role of this imaging feature in predicting therapy response in patients treated with ^223^RaCl_2_ has not been extensively studied. A previous report demonstrated a trend for shorter survival in those patients when compared to those with less than six metastases [25]; however, the relatively low frequency of this condition does not allow us to reach definite conclusions, unless large-scale studies are planned. Nonetheless, our data might suggest that some superscan patients might indeed benefit from a ^223^RaCl_2_ treatment. 

The robustness of the generated volumetric data and the clinical relevance of the information that has been derived from this analysis suggest a relevant potential for the computer-enhanced evaluation of tumor burden. The relevance of such approaches is testified to by the growing number of computer-assisted techniques, which have been developed in the last few decades to estimate the tumor burden [22]. Different approaches have been presented, for example, the bone-scan index, which is designed to be applied to planar bone scintigraphy [26], but was subject to false-positive findings in the event of a flare phenomenon [27]. Moreover, new 3D-segmentation algorithms have been introduced for PET/CT, one based on a ^18^F-NaF PET-threshold and requiring specific threshold determination for each individual scanner [28], the other using the tracer uptake of ^68^Ga-PSMA (prostate-specific membrane antigen) [29], that might be of special interest for upcoming, but currently not approved, PSMA radio-ligand therapies. The main difference of our approach lies on the use of CT-density contrast instead of the PET information for defining tumor volume as well as the applicability to any CT-based hybrid imaging, PET/CT and SPECT/CT, the latter being more widely available and less cost intensive, which is a major advantage of our current approach with SPECT/CT. Finally, the use of bone seeking tracers might better reflect the uptake of ^223^RaCl_2_ and thus predict the radiation to the metastases. One algorithm included a segmentation. 

Some limitations have to be mentioned. A possible source of error could be a misclassification of non-tumor-related bone thickening [30]. The application automatically excludes non-bone hyperdensity (e.g., vascular calcification) from the edge detection. Likewise, voxel belonging to the spinal canal and hypodense areas (such as bone cysts) are also sorted out. Our approach was shown to be congruent with the estimation of bone volume as provided by a standard commercial software. However, a comparison to an independent imaging-based reference standard provided by means of magnetic resonance or PSMA-PET/CT was not available. Moreover, comparison with other approved methods of tumor load determination, based on planar data, such as the bone scan index [31], was not possible because of their intrinsic difference. 

Another significant limitation is that in the SPECT/CT analysis, the limited resolution of the single-photon technique could underestimate the counts in smaller volumes. Finally, we evaluated the therapy response only in subjects who completed the entire ^223^RaCl_2_ therapy. This decision was made in order to have a homogenous population and to be able to perform a therapy effectiveness evaluation at the same time point after baseline staging. However, this choice excluded patients who interrupted the therapy due to tumor-progress or because of toxicity; therefore, the information presented in this manuscript applies only to the patient population with a relatively better prognosis [32]. Further studies could shed light on the correlation among tumor load, tracer distribution, and overall survival in patients with in-therapy progression.

## 4. Materials and Methods

### 4.1. Patient Population

Seventy-six consecutive patients suffering from CRPC, who underwent whole-body ^99m^Tc-bisphosphonate-SPECT/CT (mean age 69.5 ± 7, age range 55.5–80.8), were retrospectively analyzed. All examinations were performed for staging in patients with multiple bone lesions, before radionuclide therapy using ^223^RaCl2. Inclusion criteria comprised histologically confirmed prostate cancer, evidence of prostate specific antigen (PSA) increase under maximal androgen blockade, and presence of clinically symptomatic as well as radiologically confirmed osteoblastic skeletal metastases. Exclusion criteria were the presence of metal implants impeding the analysis of either the axial or the appendicular skeleton (e.g., bilateral total hip replacement, extensive spondylosyndesis, etc.), absence of a signed informed consent, and inability to complete the planned six cycles of ^223^RaCl_2_-therapy. Any previous therapy or combination of treatments was admitted. The PSA level at the time of the scan was recorded.

All patients gave written informed consent for the retrospective analysis of the pseudonymized clinical SPECT/CT data. The investigations were conducted in accordance with the Helsinki Declaration and with national regulations, after approval by the ethics committee of the University of Tübingen. All patients signed a specific consent form, detailing the use of imaging as well as of laboratory data for research purposes.

### 4.2. Patients’ Follow Up

Patients were followed up with throughout the execution of the radionuclide therapy, which included six ^223^RaCl_2_ administrations (one per month). A whole-body SPECT/CT scan was carried out at the end of therapy. This scan was re-evaluated by an experienced viewer who was blinded to the results of the computational analysis and who stratified the patients according to therapy response as follows: if new lesions were detected (whether on CT or in the SPECT images), the case was classified as progressive disease (PD). On the contrary, if no new lesions were observed, the patient was considered having a stable disease (SD). Finally, if no new lesions were observed and the uptake intensity was visibly diminished, the case was judged as partial response (PR). This study was approved by the Local Ethics Committee of the University of Tübinngen (No. 747/2017BO1) on 9 March 2015.

### 4.3. Scan Protocol

Patients were scanned on a hybrid SPECT/CT device (Discovery 670 Pro, GE Healthcare, Chicago, IL, USA), three hours after injection of 8–10 MBq/Kg of ^99m^Tc-3,3-diphosphono-1,2-propanedicarboxylic acid (CIS bio, Berlin, Germany). To minimize artifacts caused by the presence of radioactive urine in the excretory system, patients were asked to drink at least 1000 mL of water during the uptake time and to void immediately before the scan. No urinary bladder catheterization was used. 

The acquisition protocol comprised a whole-body planar scan. This part was followed by a whole-body SPECT/CT scan, from vertex up to mid-distal femur, which was obtained by reconstructing and fusing three sequential fields-of-view on a dedicated workstation (Xeleris 3^®^, GE Healthcare, Chicago, IL, USA). SPECT acquisition was carried out with the two camera heads in H-Mode; parameters for each field-of-view were as follows: energy window 140.5 ± 1 0%, angular step 6°, time per step 15″. The transaxial field-of-view and pixel size of the reconstructed SPECT images were 54 cm and 5 × 5 mm, respectively, with a matrix size of 128 × 128. SPECT raw data were reconstructed using ordered-subset expectation maximization iterative protocol (2 iterations, 10 subsets).

The technical parameters of the 16-detector row, helical CT scanner included a gantry rotation speed of 0.8 s and a table speed of 20 mm per gantry rotation. The scan was performed at 120 kV voltage and 10–80 mA current. A dose modulation system (OptiDose^®^, GE Healthcare, Chicago, IL, USA) was applied to optimize total exposure according to the patient’s body size. No contrast medium was injected.

### 4.4. Image Analysis

Segmentation of bone volumes was performed on the CT data according to the previously validated method [33,34,35]. Briefly, the algorithm identified the skeleton on CT images by assuming that compact bone is the structure with the highest X-ray attenuation coefficient in the human body. This assumption implies a stark HU value difference between soft tissue and cortical bone. The program functions by reading the HU values of every voxel in any given slice horizontally; when it encounters a sharp variation of HU density, it assumes that it had reached the bone outer border. From that point, it samples a 2-pixel ring, which corresponds to the cortical bone. It then samples the average density of this cortical bone volume. Thereafter, it categorizes every voxel on the inside of this volume as trabecular bone (bone volume, B_Vol_) or as osteoblastic metastases volume (M_Vol_). This is done by using the mean density of the cortical volume as the cutoff value, assuming the osteoblastic metastases will have an average density at least equal to that of cortical bone. Therefore, the final output of this process consists of the following volumes:-Cortical Volume (C_Vol_): the bone surface-Trabecular Volume (B_Vol_): the normal trabecular bone-Metastases Volume (M_Vol_): osteoblastic metastases (tumor burden)-Skeletal Volume (S_Vol_): entire skeletal volume (sum of C_Vol_, B_Vol_, and M_Vol_)-%INV: percent of invasion (M_Vol_/ B_Vol_ ratio)

A graphical overview of the segmentation process and an example of the program’s work on the original slices are shown in Figure 5 and Figure 6.

For details on the mathematical rationale underlying the bone recognition algorithm, please see the original work from Sambuceti et al. [33]. For an overview on the principle of tumor burden estimation, we refer to our previous work [6].

After an initial automatic segmentation, the program displayed the resulting images to the operator, who could manually exclude all benign hyperdensities (e.g., osteochondrosis, osteophytes, metal implants). Purely lytic areas, having a HU inferior to 30, were automatically removed. 

In the next step, masks corresponding to the M_Vol_ and the B_Vol_ were generated and exported onto the co-registered SPECT images; here, mean radioactivity concentration (counts/voxel) was calculated. The program’s output included the volume (in mL), the mean HU density, and the mean counts of both volumes M_Vol_ and B_Vol_. For the purpose of the present study, the skull was excluded from the analysis. Separate computations were then conducted for the whole-body skeleton (the whole skeleton from atlas until the distal femurs), the axial skeleton (vertebrae and sternum), and the appendicular bones (humeral and femoral shafts). 

### 4.5. Validation of the Computational Technique and Comparison with Controls

In order to correlate the information obtained by the new program with known indices, we compared the magnitude of volumes, density values, and tracer distribution with approved radiological, clinical, and laboratory standards of reference. In the first step, we aimed to verify the correctness of the bone identification by our program. To do so, we compared the total S_Vol_ with a volumetric estimate of the whole skeleton obtained by a licensed commercial application (QMetrix^®^, General Electric, Boston, MA, USA). This comparison was done to ensure that our method could correctly recognize the bone volume on the CT images. 

In the second step, we compared the M_Vol_ (as an estimate of the tumor burden) with the absolute number of metastatic lesions, which were manually counted on the SPECT/CT images by an expert reader. Finally, M_Vol_, mean M_Vol_ HU, average M_Vol_ counts, and ratio between M_Vol_ and B_Vol_ were correlated to PSA levels. 

To further stratify disease aggressiveness, both HU values and counts of M_Vol_ were compared between patients with or without a superscan finding at planar imaging. Superscan was defined as the presence of uniformly increased activity within the skeleton, with very faint or absent visualization of the renal system [36].

### 4.6. Statistical Analysis

Data are presented as mean ±standard deviation. The t-test for unpaired data was used to compare values between patients’ subgroups. To verify the probability of therapy response as a function of the measured counts within the segmented volumes, ROC analyses were performed, and areas-under-curve (AUC) calculated. Correlations between indices were assessed with bivariate analyses, using Pearson’s R index. Prevalence of superscan patients among groups was tested using the Chi-squared test.

A *p*-value of <0.05 was considered significant. The SPSS statistical program (SPSS^®^, v. 21.0, IBM, Armonk NY, USA) was employed. 

## 5. Conclusions

This study aimed to contribute to the transition to tailored treatments in the field of metastasized castration-resistant prostate cancer. The availability of reliable indices of disease burden and the capability to measure therapy response accurately, as well to predict its clinical course, are key in ensuring the best possible treatment to every single patient. The present paper presents a method by which disease-specific indices, mirroring the corresponding parameters of disease status, can be obtained. The capability to extract this data can potentially be used, pending further studies, to ameliorate the imaging-based disease stratification and to improve the therapeutic schedule, therefore improving treatment effectiveness in these patients.

## Figures and Tables

**Figure 1 cancers-11-00869-f001:**
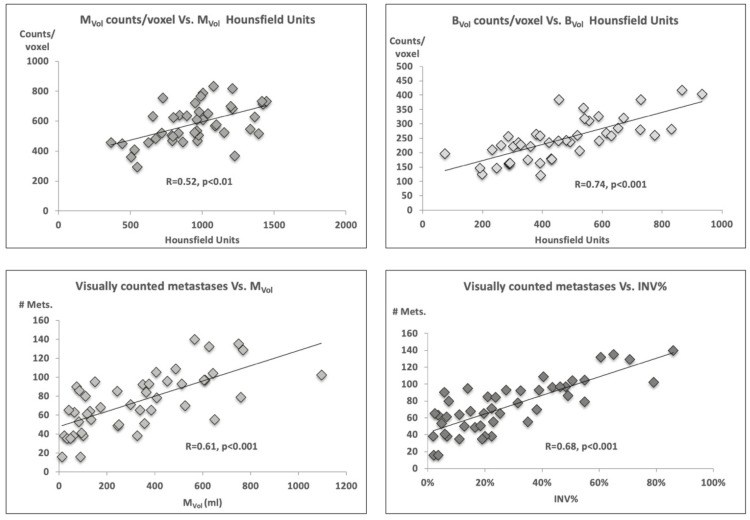
Correlation between counts/voxel and Hounsfield density as well as between volumes and number of metastases. A higher density corresponded to higher mean counts/voxel (top panels). Furthermore, a close correlation was observed between the volumetric estimates and the manual count of metastatic lesions (bottom panels). M_Vol_ = voxels metastases; B_Vol_ = normal bone; INV% = percent of bone invaded by metastases (M_Vol_/ B_Vol_).

**Figure 2 cancers-11-00869-f002:**
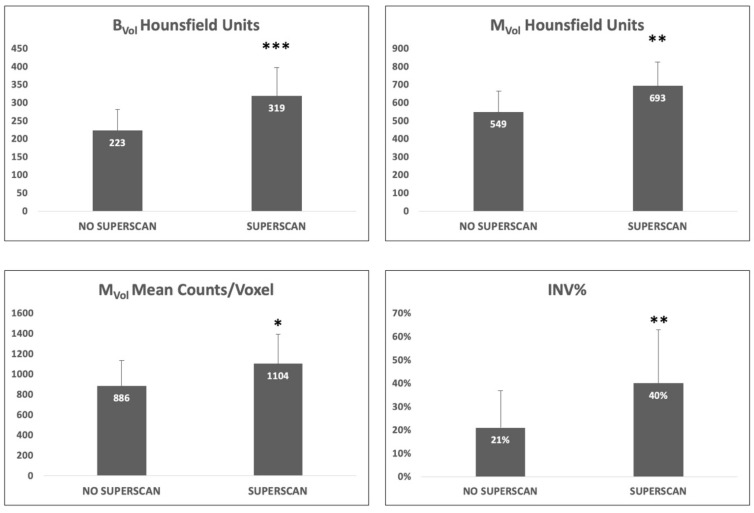
Comparison between superscan and non-superscan patients. Superscan subjects showed a higher density in trabecular bone volume (B_Vol_) as well as in metastases volume (M_Vol_) (upper panels). A higher counting rate was observed in the M_Vol_ of superscan patients (bottom left). Finally, percent of trabecular bone space invaded by metastatic lesions was higher in the superscan subjects (bottom right). * *p* < 0.05, ** *p* < 0.01, *** *p* < 0.001. INV% Percent of bone invaded by metastases (M_Vol_/ B_Vol_).

**Figure 3 cancers-11-00869-f003:**
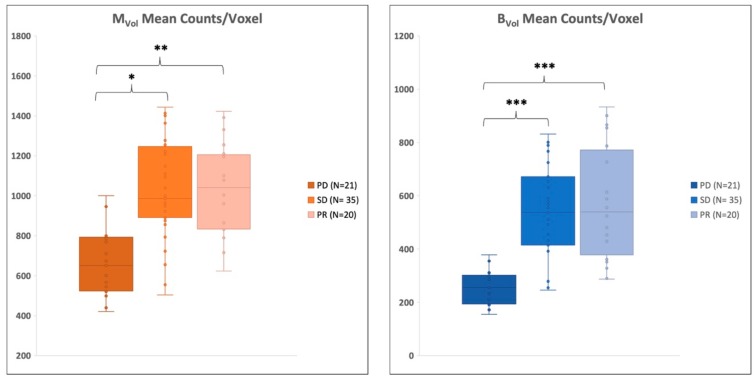
Radioactivity concentration according to response. Patients with progressive disease (PD) after the radioisotope therapy displayed a significantly lower radioactivity concentration at the baseline imaging in metastases volume (M_Vol_) as well as in trabecular bone volume (B_Vol_). * *p* < 0.05, ** *p* < 0.01, *** *p* < 0.001. SD = stable disease; PR = partial response.

**Figure 4 cancers-11-00869-f004:**
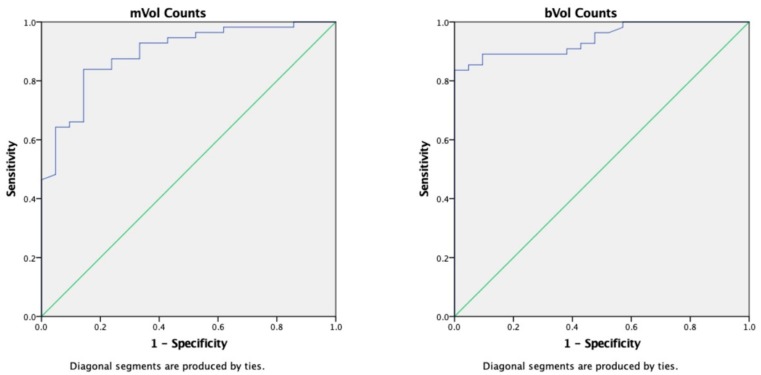
Receiver operating characteristics (ROC) curve of metastases volume (M_Vol_) and trabecular bone volume (B_Vol_) counts/voxel values according to progression. Radioactivity concentration was able to discriminate patients with a progressive disease in M_Vol_ as well as in B_Vol_.

**Figure 5 cancers-11-00869-f005:**
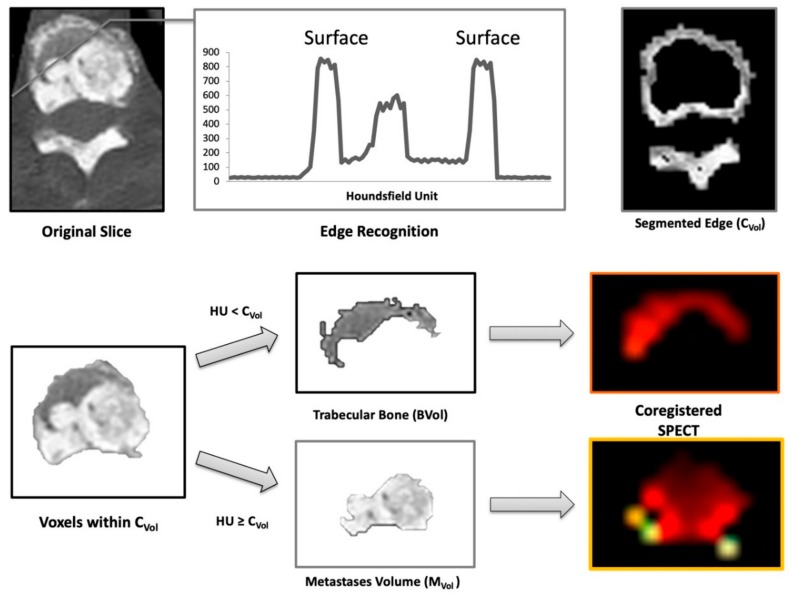
Functioning of the segmentation process. The software analyzes sequentially the Hounsfield Unit (HU) density of voxel within a single slice (top left). The bone border is identified as an increase of HU values (top center). After definition of the cortical volume (C_Vol_, top right), its mean HU density is calculated. This value is used to classify all voxels located on the inside of C_Vol_ as pertinent to bone metastases (M_Vol_) or to normal trabecular bone (B_Vol_, middle and left bottom panels). Afterwards, mean counts/voxel are extracted from the co-registered images (bottom right).

**Figure 6 cancers-11-00869-f006:**
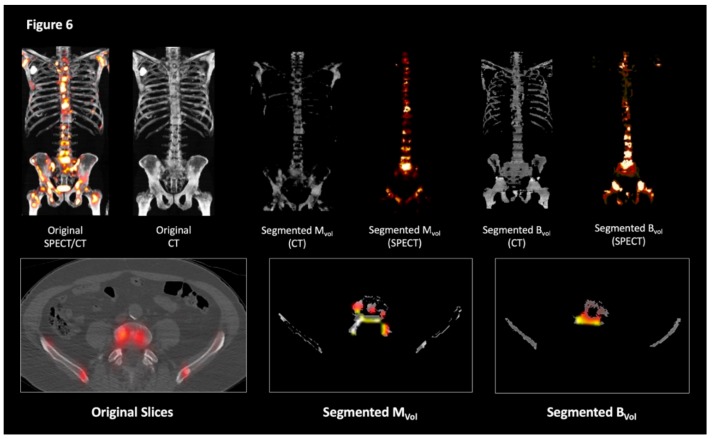
Examples of the segmentation output. Three-dimensional maximum intensity projections representations (top panels) and transaxial views (bottom panels) of the original images (left) and of the processed imaging outputs (center and right).

**Table 1 cancers-11-00869-t001:** Radiological and laboratory parameters in patients with or without superscan.

Parameter	All Patients	Superscan	Non-Superscan	*p*-Value
Mean M_Vol_ (mL)	357 ± 257	527 ± 304	245 ± 187	<0.001
Number of counted lesions	74 ± 30	106 ± 22	61 ± 23	<0.001
INV%	27 ± 20%	40 ± 23%	21 ± 16%	<0.01
PSA (ng/mL)	539 ± 754	1235 ± 959	257 ± 407	<0.001
B_Vol_ mean counts/voxel	466 ± 198	565 ± 243	428 ± 164	NS
M_Vol_ mean counts/voxel	947 ± 277	1104 ± 291	886 ± 251	<0.05
B_Vol_ mean HU	251 ± 78	319 ± 78	223 ± 59	<0.01
M_Vol_ mean HU	590 ± 136	693 ± 132	549 ± 115	<0.001

M_Vol_: Metastases Volume; INV%: Invasion% or M_Vol_/B_Vol_ ratio PSA: Prostate-specific antigen; B_Vol_: Trabecular bone volume; HU: Hounsfield Unit.

**Table 2 cancers-11-00869-t002:** Radiological and laboratory parameters according to response.

Parameter	PD	SD	PR	PD vs. PR	PD vs. SD	PR vs. SD
Mean M_Vol_ (mL)	245 ± 312	342 ± 203	373 ± 254	NS	NS	NS
Number of counted lesions	56 ± 25	80 ± 33	79 ± 26	NS	NS	NS
M_Vol_/B_Vol_	19 ± 8%	29 ± 17%	31 ± 23	NS	NS	NS
PSA (ng/mL)	353 ± 540	446 ± 539	746 ± 1004	NS	NS	NS
B_Vol_ mean counts/voxel	275 ± 60	528 ± 162	515 ± 188	<0.001	<0.001	NS
M_Vol_ mean counts/voxel	715 ± 190	1058 ± 255	975 ± 219	<0.05	<0.01	NS
B_Vol_ mean HU	232 ± 81	253 ± 68	264 ± 82	NS	NS	NS
M_Vol_ mean HU	545 ± 157	605 ± 120	610 ± 127	NS	NS	NS

PD: Progressive disease; SD: Stable disease; PR: Partial response; M_Vol_: Metastases volume; INV%: Invasion% or M_Vol_/B_Vol_; PSA: Prostate-specific antigen; B_Vol_: trabecular bone volume; HU: Hounsfield Units.

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
