# Peer review of "Automated Definition of Skeletal Disease Burden in Metastatic Prostate Carcinoma: A 3D Analysis of SPECT/CT Images"

_cancers, 2019, doi:10.3390/cancers11060869_

Round 1

Reviewer 1 Report

Dear authors,

I thank you for your interesting article. Please find hereby my comments:

1) line 23: in this study we compared the program output with existing estimate and with the clinical outcome: which clinical outcome you mean? As you defined PD, SD and PR based on radiological / imaging definitions.

2) line 220: one of exclusion criteria was presence of metal implants involving more than one third of the analyzable skeleton. Please explain the rational of "one third: ratio.

3) line 235: You define PD= new lesions, SD= no new lesions and PR= no new lesions and uptake intensity was visibly diminished. What if no new lesions were observed and uptake intensity was visibly increased?

4) line 213= Patients populations: was Biphosphonate therapy used for some patient in your selection?

5) line 257= image analysis: was DEXA scan performed for some patients in your selection? could density of osteoporotic bone influence your algorithm and results?

6)  line 64 and 73: you developed a specific computational tool, based on segmentation analysis and it works (nicely) as well as the commercial application but I see nothing about this back in your conclusion.

Reviewer 2 Report

In current study, Fiz et al., report an algorithm using SPECT/CT images from mCRPC patients to automatically detect the metastasis load on the skeleton. They also validated some results with commercial software. While this study could potentially benefit the therapy response assessment, it lacks some evidence to support the clinical application. 

1) Higher number of patients cases, and validation using more one commercial software would make the data more convincing. 

2) It is not clear how "superscan" was defined, more details should be provided on how to separate the patients.

3) It is not clear what is "above-defined criteria" in line121.

4) It is important to show the comparison by using the commercial software analysis on figure 2,3

Round 2

Reviewer 2 Report

The revised manuscript has addressed my comments.